# Mitochondria and Cancer Recurrence after Liver Transplantation—What Is the Benefit of Machine Perfusion?

**DOI:** 10.3390/ijms23179747

**Published:** 2022-08-28

**Authors:** Alessandro Parente, Mauricio Flores Carvalho, Janina Eden, Philipp Dutkowski, Andrea Schlegel

**Affiliations:** 1The Liver Unit, Queen Elizabeth University Hospital Birmingham, Birmingham B15 2GW, UK; 2Fondazione IRCCS Ca’ Granda, Ospedale Maggiore Policlinico, Centre of Preclinical Research, 20122 Milan, Italy; 3Department of Surgery and Transplantation, Swiss HPB Centre, University Hospital Zurich, 8091 Zurich, Switzerland

**Keywords:** hepatocellular carcinoma, liver transplantation, mitochondria, ischemia reperfusion injury, cancer recurrence, machine perfusion

## Abstract

Tumor recurrence after liver transplantation has been linked to multiple factors, including the recipient’s tumor burden, donor factors, and ischemia-reperfusion injury (IRI). The increasing number of livers accepted from extended criteria donors has forced the transplant community to push the development of dynamic perfusion strategies. The reason behind this progress is the urgent need to reduce the clinical consequences of IRI. Two concepts appear most beneficial and include either the avoidance of ischemia, e.g., the replacement of cold storage by machine perfusion, or secondly, an endischemic organ improvement through perfusion in the recipient center prior to implantation. While several concepts, including normothermic perfusion, were found to reduce recipient transaminase levels and early allograft dysfunction, hypothermic oxygenated perfusion also reduced IRI-associated post-transplant complications and costs. With the impact on mitochondrial injury and subsequent less IRI-inflammation, this endischemic perfusion was also found to reduce the recurrence of hepatocellular carcinoma after liver transplantation. Firstly, this article highlights the contributing factors to tumor recurrence, including the surgical and medical tissue trauma and underlying mechanisms of IRI-associated inflammation. Secondly, it focuses on the role of mitochondria and associated interventions to reduce cancer recurrence. Finally, the role of machine perfusion technology as a delivery tool and as an individual treatment is discussed together with the currently available clinical studies.

## 1. Introduction

In the United States (US), an increase in newly diagnosed liver cancers of 41‘260 and an increase in related deaths rate of 30‘520 are expected in 2022 [1]. Hepatocellular carcinoma (HCC) is the most common primary liver cancer. Liver transplantation (LT) offers an opportunity to cure, treating both the tumor and the underlying chronic liver disease. With the development of current HCC classifications and tailored listing concepts, the overall results after LT are satisfactory. In addition, LT plays an increasing role in the treatment of other primary liver cancers, including cholangiocarcinoma and colorectal and neuroendocrine liver metastases [2,3,4,5]. However, the disparity between organ utilization and demand limits the access to LT for many of those patients. To overcome such challenges, livers from extended criteria donors (ECDs), including donation after circulatory death (DCD), are increasingly utilized. Based on an evolving experience, risk-adapted organ selection is common practice to avoid well-known complications, including primary non-function (PNF), early allograft dysfunction (EAD), and ischemic cholangiopathy (IC) [6,7]. To limit the overall risk further, ECD livers are often allocated to recipients with liver tumors and preserved liver function, which may better tolerate the inflammation related with post-transplant ischemia-reperfusion injury (IRI). A thorough donor and recipient selection was suggested by several authors of retrospective cohort studies to achieve acceptable tumor recurrence rates with standard cold storage (SCS) liver preservation. However, a significant number of DCD grafts remain unutilized with this concept and novel preservation techniques being currently evaluated [8]. 

To provide uniform guidelines for donor and recipient selection and tailored preservation, a general understanding of cellular mechanisms of reperfusion injury after SCS and the role of machine perfusion is essential. Mitochondria are increasingly described as key structures in mammalian cells to cope with any type of injury. Environmental factors, drugs, hypoxia, cancer development, and ageing are some well-known features, increasingly linked to mitochondria by many. With their role to provide energy for the entire cellular metabolism, mitochondria are also known as “power houses” and are key to every subcellular function and decisive for the survival of an organism after a specific injury (Figure 1).

When a donor is available, a certain level of tissue hypoxia always occurs and triggers an overall impaired metabolism. The mitochondrial metabolism is always impaired to a certain extent and prompts the well-known cascade of ischemia-reperfusion injury (IRI) when oxygen becomes available again (reperfusion). The main feature is a pro-inflammatory cascade, which leads to more or less clinical post-transplant complications based on the overall organ quality and function. One direct consequence after reperfusion is the development of a pro-tumor tissue milieu, which enables the migration and regrowth of circulating tumor cells (CTCs) with subsequent cancer recurrence. Machine perfusion (MP) with the overall aim to reduce IRI-features is currently tested as a promising tool to protect recipients from various post-reperfusion and IRI-associated complications. The evaluation of a potential impact on liver cancer recurrence appears as logical consequence. This review article therefore provides mechanistic insights into the IRI cascade and the role of mitochondria and their link to cancer recurrence. An additional focus is on current interventions targeting mitochondria, including machine perfusion, to reduce tumor recurrence and the development of metastases after liver transplantation. 

## 2. Mechanisms of Ischemia-Reperfusion Injury

Organ injury starts already in the donor with episodes of hypoxia and hypoperfusion during rescue treatment to save the donor, e.g., during prolonged intensive care unit admissions, when most brain functions are lost. The better the tissue can cope with such effects, the lower the later inflammatory injury after reoxygenation will be. During and after procurement, livers are exposed to additional periods of warm and cold ischemia. 

The ischemic injury starts in the donor when oxygenated blood flow ceases, which corresponds to the time of cold organ flush in brain-dead donors (DBD) and to the time introduced with relevant hypotension and hypoxia during DCD donation (e.g., donor warm ischemia time) [9]. Cellular hypoxia with inhibition of electron flow through the respiratory chain and subsequent ATP-loss appears as immediate consequence together with an accumulation of toxic metabolites and ions, including Na^+^, K^+,^ and Ca^++^, due to the failure of ATP-dependent ion channels. Particularly, a prolonged ischemia causes cellular edema, which activates proteases and initiates the cascade of apoptosis [10]. Of great importance is the accumulation of Krebs-cell metabolites, including succinate during hypoxia [11], which trigger the immediate release of reactive oxygen species (ROS) from complex I, when oxygen is reintroduced [11,12]. With the ROS-release in the first few minutes after reperfusion, further pro-inflammatory molecules, including Danger-associated molecular patters (Damps) [13], are also released from severely injured cells, including hepatocytes, and are recognized by activated Kupffer and sinusoidal endothelial cells (SECs). Damps molecules trigger a sterile inflammation through further secretion of pro-inflammatory cytokines (e.g., TNF-α, IL-1b, IL-18) [14]. Damps together with ROS release can activate two pathways: first, the transcription and release of increased levels of pro-cytokines from the nucleus (e.g., IL-1β), and second, the assembling of the inflammasome (NLRP-3) and capsase-1 cleavage to finally activate and cleave the pro-cytokines [14]. Damps receptors are toll-like receptors, which can contribute to trigger the activation of the transcription molecules AP-1, NFκB, and IRF3. Such factors promote the maturation of antigen presenting cells (APCs), including dendritic cells, which present antigens and express co-stimulatory molecules. The direct consequence is the increased inflammatory tissue response with the activation of T cells and other immune cells, leading to additional ROS release with downstream injury of initially healthy cells [14].

This acute and ongoing inflammation also involves further cellular and subcellular compounds including complement proteins (e.g., C3a and C5a), which become activated and recruit circulating recipient neutrophils, which, in turn, migrate directly into the newly implanted liver, amplifying the downstream activation of the innate immune system with further impaired cellular functions and death [15]. Another element is the activation of platelets with the formation of micro-thrombi inside the hepatic sinusoids, which impair the liver microcirculation further, thereby aggravating hypoxia and hypoperfusion [16].

Nowadays, there is mounting evidence demonstrating that mitochondria play an additional key role in oncogenesis. In fact, the ROS produced by mitochondrial metabolism can damage cells and mitochondrial DNA, which can result in new mutations, and upregulate proto-oncogenes together with a downregulation of tumor-suppressor genes [17]. However, the link between regional liver-IRI and systemic effects creating the perfect milieu for existing cancer cells to resettle and replicate is of increasing interest. 

## 3. Specific Mechanisms of Cancer Recurrence after Liver Transplantation

Any intervention with subsequent tissue trauma, including surgery, may trigger inflammation and hypoxia. Organ procurement and transplantation is no exception. The overall aim should, therefore, be considered as the limitation of injury, both surgical and medical (Figure 2). The increasing body of literature demonstrating the link between the level of injury (e.g., organ quality and level of recipient disease), the mitochondrial response during reperfusion, and further downstream effects with organ dysfunction and complications leads the way to understanding how to prevent this cascade. The physiological stress related to the surgery itself also plays an important role [18]. 

With an experienced donor and recipient surgical team, the trauma can be reduced, further supported by an efficient anesthesiologic and the medical management of both donor and recipient [19,20]. Donor type, graft quality, and preservation contribute significantly to the overall IRI-level after reperfusion. Livers with prolonged warm and cold ischemia, combined with other risk factors, including macrosteatosis, are at particular risk. Such combinations induce the highest levels of ROS, Damps, and cytokines with the subsequent activation of the innate immune system with a general inflammation and downstream complications after liver transplantation [21]. 

Additional downstream consequences become visible within the micro-environment of the newly implanted liver, and also systemically. With the size of an adult liver, a large number of molecules are released into the recipient’s circulation after reperfusion, leading to a systemic inflammatory response. Straightforward transplant procedures (donor and recipient surgery) of low or standard liver grafts, which become implanted in low lab MELD candidates with medical stability, will have less inflammatory features with proper immediate multi-organ function, when compared to constellations with a higher risk. 

### 3.1. The Contribution of Mitochondria

Mitochondria were identified as key contributors, which trigger overall tissue inflammation and thereby create an attractive environment for CTCs to resettle and grow. The ROS–Damps–cytokine cascade instigates further downstream effects in the liver micro-environment, enabling CTC migration through the SEC barrier. This cascade is of high relevance in transplantation with higher donor and recipient risk and in the context of the additional risk factors described before.

Mitochondria further contribute to cancer recurrence and new development through various other pathways, for example, with mitochondrial DNA mutations [22]. ROS molecules are also known to be involved in cancer progression, and high ROS levels are often found in cancer cells [23]. Mitochondria further regulate cell death and apoptosis, control the metabolic reprogramming through genetic mutations, encode tricarboxylic acid cycle (TCA) metabolites, and sustain cellular proliferation. Such features are generally linked to tumor generation, progression, and metastases [24]. 

Another key role of mitochondria is described for the process of metastatic dissemination [25]. The first step is the epithelial–mesenchymal transition (EMT), promoted by the mitochondrial biogenesis and oxidative phosphorylation (OXPHOS) [26]. The downregulation of OXPHOS was found in several tumor types, and is based on mutations seen in the mitochondrial DNA. At the same time, the glycolytic pathway was frequently upregulated [27]. Tumor energy is mainly derived from glycolysis; however, in cancer cells, the TCA cycle is often maintained in an active state to guarantee the availability of anabolic substrates and oncometabolites, which support neoplastic proliferation. Another important mitochondria-related pathway, with a link to cancer, is the biosynthesis of heme. Targeting OXPHOS, heme metabolism in mitochondria can serve as a compelling object for novel therapies to treat cancer [28]. Mitochondrial biogenesis and turnover is another potential target for new anti-cancer interventions [28]. 

### 3.2. The Complex Interplay between Liver Micro-Environment and Immune System 

Related with numerous pathways present during early IRI, a variety of immune cells play a central role in cancer development and progression. The link between inflammation and cancer succession is well-established and mediators released by cells in the tumor micro-environment (TME) contribute significantly [14,29]. Understanding the TME is, therefore, of importance to identify strategies to reduce the HCC recurrence after LT. IRI-based inflammation triggers local changes with the development of a favorable micro-environment for tumor cells to invade, migrate, and grow [30,31]. 

Hepatic sinusoidal endothelial cells (SEC) play a key role here, with their direct response to IRI and the swelling and subsequent loss of integrity with increased micro-vascular permeability, which facilitates tumor cell permeation. In fact, hypoxia stimulates the secretion of hypoxia-inducible factor (HIF-1α), which promotes tumor cell proliferation, migration, and angiogenesis, with the secretion of vascular endothelial growth factors (VEGF) [31,32,33,34]. 

Several other pathways are also linked to IRI and tumor recurrence. The release of ROS and Damps triggers a high concentration of inflammatory cytokines in the circulation. Such molecules, including TNF-α and IL-1, play an important role for tumor progression because their upregulation triggers the expression of adhesion molecules (e.g., e-selectin and vascular cell adhesion molecule-1) on endothelial cells, which act as mediators for tumor growth [33,34]. 

Figure 3 describes some of the local and systemic effects of IRI-associated effects on the micro-environment after liver transplantation and subsequent HCC recurrence. 

An activated IRI cascade also promotes tumor cell invasion and migration through an upregulation of multiple Rho-family molecules, which are crucial for cellular motility, proliferation, and apoptosis. Their overexpression is linked to tumor infiltration and metastasis in organs affected by hepatic IRI [35]. Furthermore, IRI promotes the release of CXCL-10, a chemoattractant, which increases the recruitment and differentiation of endothelial progenitor cells (EPC) and macrophages towards the liver. This augments tumor cell invasion and infiltration into blood vessels [36]. The CXCL-10 activation pathway is further associated with neo-angiogenesis and was shown to enhance the tumor cell motility and promote the development of more invasive phenotypes [37,38]. Additional support for this mechanism comes from authors who describe the potential role of circulating EPCs as prognostic marker in HCC patients [39]. 

Activated through the ROS-Damps axis, APCs including macrophages and dendritic cells, cross-present antigens on major histocompatibility complex (MHC) class-I molecules to CD^8+^ T cells, using the T-cell receptor (TCR). Subsequently, naïve CD^8+^ T cells turn into cytotoxic effector T lymphocytes (CTLs). This mechanism is crucial in cancer immunology because CTLs can detect tumor antigens using TCRs, thus destroying tumor cells [40,41]. In HCC, the presence of a high number of CD^8+^ T cells was associated with a better prognosis [42]. 

APCs also secrete stimulating factors, including VEGF and matrix metalloproteinases (MMPs), thereby promoting tumor cell growth and dissemination [43]. In addition, endothelial cell activation occurs, which leads to the recruitment of fibroblasts and mesenchymal stem cells, which, in turn, provoke the release of soluble growth factors with subsequent cancer cell progression [44]. 

Circulating pro-inflammatory mediators, including prostaglandin E_2_ (PGE_2_), represent an additional response to the surgical trauma and stimulate cancer-promoting regulatory T cells, the reduction in activated CD^8+^ T cells, and induce a shift from anti-tumor T helper 1 (TH_1_) cells towards tumor-promoting TH_2_ cells [45,46].

Thereafter, tumor-associated macrophages (TAMs) reside in the TME and promote the regional angiogenesis together with tumor progression and metastasis [47]. There is experimental evidence showing that TAMs play an additional role with their direct effect on tumor cells and also through immune system modulation with cytokine secretion (e.g., TNF- α, IL-6, and IL-1β) [48,49]. 

Further research demonstrated that CXCL-10 interact with CXC-chemokine receptor-3 (CXCR-3), which upregulates the recruitment of regulatory T cells (Tregs) within the liver [50]. Tregs are a subset of CD^4+^ T cells, characterized by an expression of Foxp3, a protein involved in the immune system response [51]. Their main function is to maintain self-tolerance and to prevent auto-immunity response. Tregs secrete various interleukins (IL-10, IL-35, TGF-β, IDO, VEGF), which induce T-cell suppression. For example, IL-35 facilitates intra-tumoral T-cell exhaustion through the expression of multiple inhibitory receptors (PD-1, TIM-3, and LAG-3, or the BLIMP1-inhibitory receptor axis) in combination with IL-10 [51,52]. Tregs from HCC patients were shown to decrease cellular proliferation, activation, degranulation, and the production of granzyme A, granzyme B, and perforin in CD^8+^ T cells [53]. Tregs also promote angiogenesis and downregulate the expression of co-stimulatory molecules CD^80^/CD^86^ on dendritic cells with subsequent suppression of their inherent cytokine production, including lower TNF-α and IL-12 release [54]. A recent study has linked high levels of Tregs with multiple HCC nodules, poor histological differentiation, higher alpha fetoprotein (AFP) levels, and vascular invasion leading to poorer survival rates [55].

## 4. Risk Factors for Liver Cancer Recurrence after Transplantation

Although LT offers a curative option for HCC, tumor recurrence remains a concern. The current literature demonstrates a tumor recurrence between 2.8% and 15.3% or even 23.6% when considering different graft types and follow-up durations [8,56,57]. HCC recurrence rates are related to the graft quality and the sub-clinical, unknown extrahepatic spread of the tumor at the time of LT and/or nesting of tumor cells during the manipulation of the liver during transplant surgery. A detailed understanding of tumor characteristics and thorough recipient selection are paramount to obtaining satisfactory results. Although the Milan criteria promote excellent survival rates after LT, they are solely based on the tumor morphology [58]. In the last decades, the understanding of other important biological factors, including total tumor volume and AFP, led to an expansion of the Milan criteria [44,59,60]. Additional parameters beyond standard tumor characteristics, such as donor and graft quality and recipient immunosuppression, were identified as key players. Orci et al. demonstrated that patients who received a liver from elderly donors (>60 y) or with diabetes or a body mass index (BMI) of ≥35 kg/m^2^ had a higher post-transplant recurrence [61]. Advanced donor age was also found to promote HCC recurrence in other studies, but the results appear controversial [62,63,64].

Based on their significant contribution to the IRI cascade, livers from ECDs were shown to induce higher complication rates after transplantation [65]. Increased risk is also observed with steatotic livers, which accumulate more succinate and NADH during ischemia and subsequently generate more ROS molecules during early reperfusion due to their fat metabolism [66,67]. Similar features were described for DCD livers, where more ROS and other inflammatory molecules are generated [8,68,69,70,71]. Prolonged donor WIT and CIT were both independently associated with a higher risk for HCC recurrence in multiple studies [56,57]. With DCD grafts in particular, it was shown that a prolonged CIT of >8 h is related with poor outcomes [72]. Other studies nominated an advanced donor age of >60 y, prolonged CIT of >6 h or 10 h, and a prolonged graft implantation time of >50 min as key risk factors [8,56,57,61,68,69,70,71,73]. The overall literature is conflicting, with some studies relating DCD grafts with a higher HCC recurrence compared to the use of DBD grafts, while others demonstrate comparable results.

The duration of donor WIT may certainly play a key role. Livers from donors can, however, also entail a higher risk based on an additional prolonged CIT with subsequent higher tumor recurrence rates. Unsurprisingly, elevated peak liver enzymes after LT from ECD donors were associated with increased HCC recurrence risk in candidates within the Milan criteria [73]. The currently available studies focusing on risk factors associated with IRI and HCC recurrence are summarized in Table 1.

## 5. Strategies with the Potential to Reduce Cancer Recurrence

The option to select and discard certain donors may reduce HCC recurrence, but it will, however, not solve the lack of donor organs for the high number of candidates. The reduction in donor WIT and CIT is routine today, particularly when other risk factors are present. This is of interest as it is a low-cost measure and a widely applicable principle with which to effectively minimize preservation injury. However, multiple other strategies were studied to limit IRI, which include pharmacological interventions, surgical procedures with hepatic inflow modulation, and the introduction of dynamic organ preservation technologies.

### 5.1. Pharmacological Agents

Anesthetic drugs were previously used to modulate IRI, both in animal models [74] and in randomized controlled trials (RCTs) for patients undergoing liver surgery [75,76]. The results are satisfactory, with a significant reduction in IRI-related features. However, there is a lack of studies regarding the utilization of these agents in the setting of LT. The antibiotic molecule rifaximin was described to modulate the intestinal microbiota, which is thought to contribute to IRI. Authors have demonstrated a rifaximin course of <28 days as an independent risk factor for graft dysfunction, suggesting a therapeutic role of rifaximin [77].

N-acetylcysteine (NAC) was evaluated in an RCT to prevent IRI and was associated with significantly higher concentrations of both anti-inflammatory interleukins before (IL-4, IL-10) and after reperfusion (IL-4), which may have an attenuating IRI-effect [78].

Corticosteroids are common therapeutic measures used to treat inflammation and their role in reducing IRI was evaluated; however, no well-established effects on the liver were reported [79]. Although pharmacological agents seem to reduce IRI, there is no report linking their application with tumor recurrence to date.

### 5.2. Tailored Immunosuppression 

One of the main factors to prevent the progression of cancer cells is immune defense. However, tumors can escape from the immune surveillance by creating an immunosuppressive environment. In this context, medical immunosuppression (IS) and particularly the group of calcineurin inhibitors (CNIs), despite preventing graft rejection, are key targets here. The relation between IS and cancer development is well-established [80]. Looking at HCC recurrence, CNIs were linked to a higher post-transplant tumor recurrence with a dose-dependent association [81,82]. Conversely, other IS regimens, which are based on the mammalian target of rapamycin (mTOR) inhibitors, including sirolimus and everolimus, were found to reduce HCC growth [83]. The association between mTOR inhibition and HCC recurrence after LT was investigated in various studies with controversial results [84,85,86]. 

There is currently no standardized protocol for the management of liver recipients with HCC, although the type and level of immunosuppressive drugs play a key role [81]. Based on the long-term side effects of IS, minimization strategies were explored and justified by intrinsic liver tolerance, which allows IS tapering without directly compromising patient and graft survival [87,88]. However, the impact of IS tapering on HCC recurrence was only investigated in a single, retrospective study [89] and, therefore, is still under debate, although a few practical guidelines were developed [90]. An Italian working group suggested pathways for the post-transplant immunosuppression, which consider the primary liver disease, the medical recipient status, early post-operative events, and expected complications, including nephrotoxicity and de novo malignancies. Regarding the management of patients undergoing LT for HCC, the authors recommended steroid-free immunosuppression and the reduction in CNI exposure by introducing mTOR inhibitors, given their anti-tumor potential [84,85,86,90].

### 5.3. Surgical Interventions

The impact of surgical interventions was explored in the context of IRI after transplantation. The Pringle Maneuver is the most common mechanical procedure to occlude liver inflow for a short period followed by repeat reperfusion (“occlusion-release-cycles”). This technique is also called ischemic preconditioning (IP) and was shown to have beneficial effects, with a reduction in IRI-associated inflammation and HCC progression. Indeed, some authors demonstrated that a few short IP cycles reduced the overall tumor burden in ischemic steatotic livers to the level of healthy steatotic controls [91]. Some human studies have demonstrated the beneficial effect of IP in donor livers before transplantation and demonstrate lower peak transaminases and a lower PNF incidence [92,93]. Such findings were confirmed in a meta-analysis in which lower PNF- and better patient survival rates were observed in livers with IP before transplantation [94]. A certain number of short cycles with in situ ischemia (e.g., IP) with subsequent “normothermic reperfusion” seem to upregulate inherent cellular defense mechanisms, such as the HIF-1alpha pathway, with subsequent liver protection after subsequent implantation [95]. Additional protection was seen with the induction of limited IRI-associated inflammation with combined ischemic pre- and post-conditioning [57,96]. 

Another technique was described with remote ischemic preconditioning (RIPC), where the intermittent pneumatic peripheral compression of a limb was found to upregulate the defense mechanisms related to platelets and the serotonin–VEGF–MMP8 axis [97]. The majority of these techniques require further studies with long-term follow-up to link the known protection from IRI with a lower tumor recurrence rate. 

### 5.4. The Role of Organ Preservation Strategies

With the given donor and recipient risk parameters, organ preservation seems the most attractive approach to improve results after LT and is, therefore, currently a hot topic in our field. Machine perfusion offers the opportunity to improve organ function and test viability before transplantation [98]. Two main concepts are explored today. First, in situ perfusion where, for example, organs in the abdominal compartment undergo normothermic regional perfusion (NRP) immediately after a period of donor WIT [99]. The results from non-randomized trials appear encouraging, with satisfactory short-term outcomes after transplantation of DCD livers procured with NRP [99,100,101,102,103]. This technique is routinely used to procure DCD livers in Spain, France, Italy, and in some regions in the United Kingdom (UK) [104]. Another concept emerges with the ex-situ perfusion of routinely procured organs, either starting at the donor site and throughout transport or in the recipient center following a period of SCS. 

While all techniques are performed under various conditions, including different perfusion timings (e.g., at donor hospital, during organ transport, and/or pre-implantation) and temperatures, the common goal is to reintroduce oxygen into hypoxic tissues as soon as possible [105,106]. Such ex-situ techniques are either performed under normothermic conditions with the aim to achieve a near-physiological environment using blood-based perfusate or under hypothermic conditions. The cold technique was first used in clinical practice in 2010 by Guarrera et al. [107]. The key is a high oxygen concentration of >60 kPa in the perfusate to trigger the typical reprogramming and repair of mitochondria before rewarming. Such hypothermic oxygenated perfusion (HOPE) re-establishes a forward electron flow with the functioning aerobic metabolisms and subsequent ATP recharging, and a reduction in previously accumulated toxic metabolites, including NADH and succinate [12]. At subsequent normothermic reperfusion, ROS release is significantly lower compared to direct normothermic reperfusion, with lower IRI-features and complications. Due to this mitochondrial protection, downstream inflammation is reduced and other cellular structures are protected, including the nucleus [108]. In addition to the protection of hepatocytes, the activation of other liver cells is reduced, including Kupffer and endothelial cells. Importantly, the micro-environment of the liver appears less inflamed and is, therefore, less attractive for CTCs to resettle and regrow. Livers after HOPE treatment induce a significantly reduced immune response [109]. Of note, this was also seen in kidneys [110]. Related to this reduced inflammation, the HOPE procedure further diminishes later biliary complications based on a lower release of profibrotic molecules from stellate cells and myofibroblasts after LT [111,112]. Whilst these findings were mainly explored in experimental and clinical studies with DCD livers, there is also some evidence in DBD grafts from elderly donors and in fatty livers, where HOPE was found to reduce IRI with subsequent improved early function and graft survival [67,113,114].

The impact of HOPE on clinical results was investigated in many studies, which consistently show the protective effects, with reduced IRI, less bile duct injury, and better graft function, with a follow-up of up to 5 years [107,114,115,116,117,118]. Additional evidence comes from three RCTs, which confirmed the HOPE effect, with protection from post-transplant biliary complications, EAD, high peak transaminases, and better graft survival rates in various graft types, including DCD livers [119,120,121]. While further studies on the optimal perfusion timing and duration are ongoing, recent evidence from Europe demonstrates the safe facilitation of transplantation logistics and a prolonged preservation period with cold perfusion [122]. 

In addition, the analysis of HOPE perfusates also allows for the assessment of liver viability, because when oxygen is reintroduced after ischemia, various molecules are released from mitochondrial complex proteins, including flavin mononucleotide (FMN), a marker with great potential to predict liver function during machine perfusion [12,123]. Perfusate FMN levels were associated with later graft survival and complication after LT. 

In contrast, ex situ normothermic machine perfusion (NMP) techniques are routinely performed with oxygenated blood at 37 °C [124,125,126]. During NMP, the liver is metabolically active, which may offer another opportunity to assess viability, for example, through biochemical perfusate and bile analyses, including transaminases, glucose, lactate, bile volume, and pH [127,128,129]. Two RCTs have explored the role of NMP in clinical practice and demonstrated the effective reduction in post-transplant recipient transaminases with subsequently lower EAD rates [130,131]. When performed instead of cold storage during transport, NMP reduces the release of pro-inflammatory molecules and upregulates genes of tissue regeneration and platelet function [132]. 

Of note, the most efficient NMP technique is the labor-intensive version of “ischemia-free” organ transplantation (IFOT), which is currently being explored by colleagues from China. The perfusion device is connected to the liver in the donor and entirely bridges the time between donor and recipient, where the organ is disconnected and implanted with overall minimal warm ischemia and the avoidance of cold flush and cold storage [106]. In contrast, the use of NMP after relevant SCS leads to a comparable inflammation as that seen with cold storage preservation alone, including similar biliary complication rates with the use of DCD livers [129,133]. The overall landscape of studies with machine perfusion confirms the underlying mechanisms of IRI induction when oxygen is reintroduced under normothermic conditions. This perfusion technique may, however, work well in cases with limited injury, such as short warm or cold ischemia times [130,133,134]. This was recently confirmed with the achievement of a prolonged 3-day NMP of a low-risk human liver with subsequent implantation [135]. 

## 6. The Impact of Machine Liver Perfusion on Tumor Recurrence

Based on previously described concepts of IRI-related inflammation with an impact on the liver micro-environment and post-transplant complications including cancer recurrence, machine perfusion appears to be attractive tool to improve current results. The protective effect of HOPE was also recently demonstrated in the context of liver cancer. A retrospective matched cohort study from two centers in the UK and Switzerland assessed the outcome after transplantation of candidates with HCC. Recipients of cold-stored DBD livers experienced 4-times higher HCC recurrences rates compared to high-risk DCD liver recipients that underwent endischemic HOPE treatment prior to implantation [136]. Of note, the recipient tumor burden was relatively high with 20–30% of candidates outside the standard HCC criteria, including Milan, UCSF, and Metroticket 2.0 [136]. These results demonstrate that a short mitochondrial treatment with HOPE before normothermic reperfusion is key to limit the inflammatory response and provide an anti-cancer effect (Figure 4). 

In contrast, the IFOT approach, could be a feasible technique in organs from extended DBD donors. Interestingly, one study demonstrated the protective effect of IFOT in recipients with HCC. Lower recurrence rates were seen through an IRI reduction [137]. In this propensity score matched analysis, the authors illustrated a better recurrence-free survival after IFOT (HR 3.728, 95% CI 1.172–11.861, *p* = 0.026), when compared to SCS alone [137]. The authors showed recurrence-free survival rates at 1 and 3 years after LT in recipients with HCC in the IFOT group of 92% and 87%, respectively, which were significantly higher than those (73.0% and 46.3%) seen in the cold storage group [137]. Looking at the HCC burden, in the entire cohort, the pre-transplant AFP level was higher in the non-IFOT group compared to the IFOT group (*p* = 0.016). The percentage of LT recipients within the Milan criteria and micro-vascular invasion were also higher in the non-IFOT group than in the IFOT group (*p* = 0.032 and *p* = 0.042, respectively). There were no differences in the size of the largest HCC lesion, the number of lesions, tumor differentiation, or the type of previous treatment between the two groups [137]. The two clinical studies with the impact of HOPE and IFOT are summarized in Table 2.

Although there is some evidence to demonstrate that the underlying liver disease could play a role in the development of post-LT HCC recurrence, especially a hepatitis B and C infection [138,139], the effect of novel perfusion technologies in this context remains unclear due to a lack of available data. Further studies to identify the subset of grafts and recipients who would benefit most from MP in the context of the underlying liver disease are required.

## 7. What Is the Potential Impact of IRI on the Recurrence of Other Liver Tumors and Metastases?

In the last decade, LT was explored as a therapeutic option for other types of primary and secondary liver tumors, including cholangiocarcinoma (CCA), neuroendocrine (NETs), and colorectal liver metastasis (CLRM) [5,140]. It is known that the immune system plays a key role in the context of the progression and recurrence of non-HCC liver tumors. Indeed, a suppressive immunologic TME plays a tumor-promoting role in CRLM, which is related to TAMs and Tregs. However, TAMs maintain the immunosuppressive environment by expressing checkpoint-ligand-programmed-death-ligand-1 (PDL1), PDL2, and other inhibitory receptors and activate Tregs by secreting IL-10 and tumor growing factor (TGF)-β. TAMs also release multiple other molecules, which remodel the extracellular matrix (ECM), including the plasminogen activation system, MMPs, and kallikrein-related peptidases. Such factors augment the migration of tumor cells. In addition, when targeting the CCL2/CCR2 chemokine axis, TAMs infiltration at the metastatic site is reduced and metastatic colorectal cancer cells are sensitized to tumor T cells [141]. 

Patients affected by CCA often present with cholestasis, which has been linked to increased IRI and associated damage [142]. In the development of CCA, several molecular and genetic pathways within the hepatic TME have been demonstrated. Importantly, necroptosis, a recently discovered process of regulated cell death, was found to contribute to CCA growth [143]. Damps molecules, released by necroptotic hepatocytes, can activate the immune response with the induction of a pro-inflammatory environment, determining the outgrowth of CCA from transformed hepatocytes. Epigenome and transcriptome profiling of mouse HCC and ICC singled out Tbx3 and Prdm5 as the main micro-environment-dependent and epigenetically regulated lineage-commitment factors, a function that is conserved in humans [143]. Several other described mechanisms, including the HIF-1α-related pathways, promote CCA progression and the development of metastases [144]. Such molecular mechanisms are potentially targeted by HOPE and IFOT through the reduction in mitochondria-associated IRI features and the subsequent limited activation of the innate immune system. With more evidence in the future, LT could be a well-defined approach to treat CCA, metastatic NETs, and CRLM.

In this setting, the modulation of inflammation and the subsequent reduction in tumor recurrence appears to be of great importance. Future studies should, therefore, also assess the role of MP on outcomes after transplantation in recipients with non-HCC liver tumors and metastases.

## 8. Summary and Future Perspective

Based on the underlying mechanisms of cancer recurrence, the importance of the TME, and the role of mitochondria, it appears that one effective strategy to prevent HCC recurrence after LT is the reduction in IRI. Nowadays, ECDs are being utilized more frequently and their vulnerability to elevated IRI warrants careful decision making when accepting such organs. In this context, dynamic perfusion strategies, which improve mitochondrial metabolism and reduce IRI, are of great interest. Further studies, including large prospective trials, are needed to confirm these results and thus change the current standard preservation techniques and improve future outcomes. With the increasing acceptance of candidates with liver metastases and liver cancer types, other than HCCs, frequently transplanted with marginal grafts, the concept of IRI reduction deserves recognition and should be further explored in the future. Equally, the impact of early liver function, inflammation, and recipient recovery together with the maintenance of mitochondrial health may play a key role in the development of secondary cancers after liver transplantation. More experimental and large clinical cohort studies with long-term follow-ups are required to provide further details regarding the potential role of new preservation strategies in the context of tumor recurrence and secondary cancer development. 

## Figures and Tables

**Figure 1 ijms-23-09747-f001:**
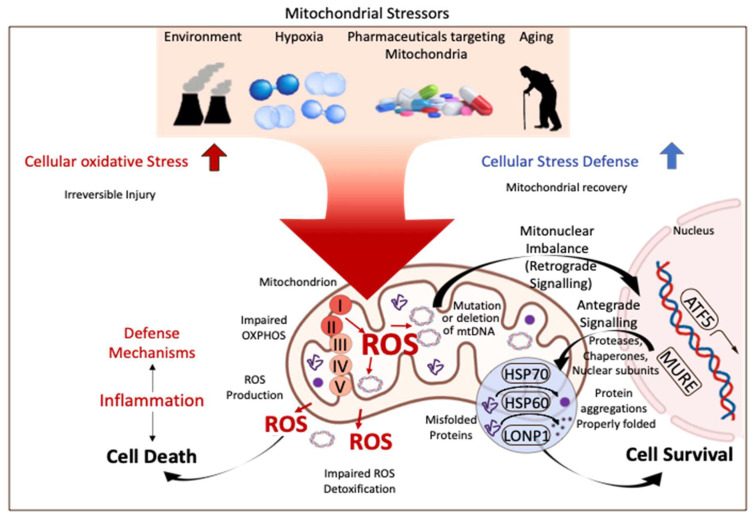
Mitochondrial stress response and signaling in injury and disease. Mitochondria are key compounds in all cells and balance injury with resilience to defend cells from stress induced by the environment, related to ageing, or from hypoxia. Mitochondria are therefore crucial for cells and organs to survive the process of donation, preservation, and transplantation. ROS: reactive oxygen species; ATF 5: activating transcription factor; HSP 60/70: heparan sulfate 60/70; LONP1: Lon peptidase 1. This figure was designed with the support from biorender.com (accessed on 15 June 2022).

**Figure 2 ijms-23-09747-f002:**
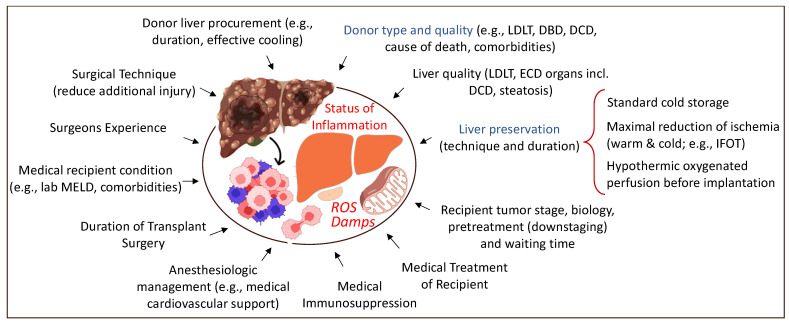
Overview of tributary factors to tumor recurrence in the context of liver transplantation. Various surgical and medical elements contribute to a higher local and systemic inflammation with frequent tumor recurrence. LDLT: living-donor liver transplantation; DBD: donation after brain death; DCD: donation after circulatory death; ECD: extended criteria donor; MELD: model of end-stage liver disease; IFOT: ischemia-free organ transplantation. This figure was designed with the support from biorender.com (accessed on 15 June 2022).

**Figure 3 ijms-23-09747-f003:**
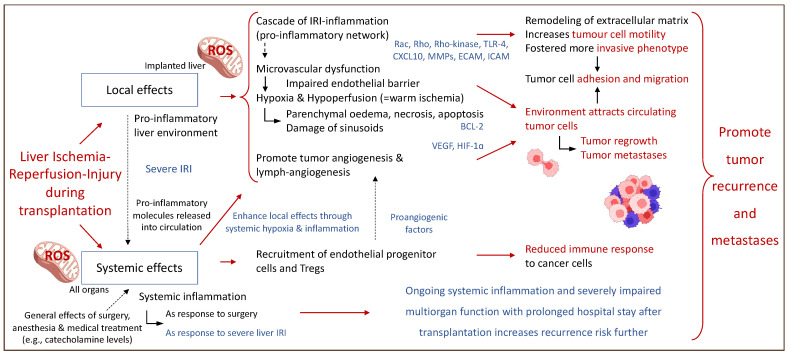
Perioperative events with an impact on the fate of residual cancer cells after liver transplantation. Local and systemic effects establish a complex interplay and contribute to cancer recurrence involving the entire system of the liver recipient. Rac: Rho family of nucleotide guanosine triphosphate hydrolase enzymes; Rho: Rho family of nucleotide guanosine triphosphate hydrolase enzymes; TLR-4: toll-like receptor-4; CXCL10: C-X-C motif chemokine ligand; MMPs: metalloproteinase; ECAM: epithelial cell adhesion molecule; ICAM: intercellular adhesion molecules; BCL-2: B-cell lymphoma 2 gene; VEGF: vascular endothelial growth fact; HIF-1∝: hypoxia-inducible factor 1∝. This figure was designed with the support from biorender.com (accessed on 15 June 2022).

**Figure 4 ijms-23-09747-f004:**
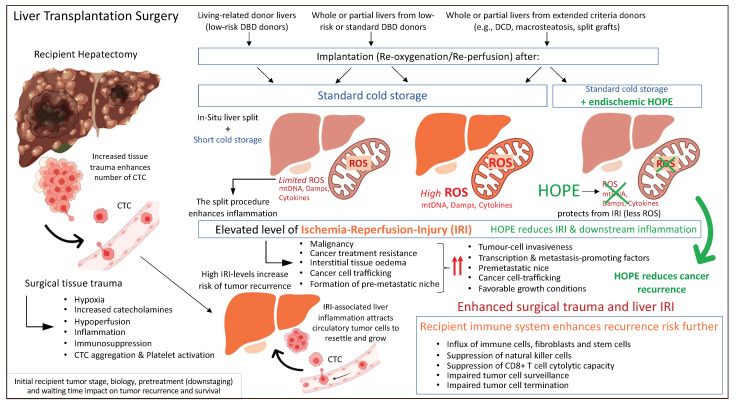
Peri-transplant events with an impact on residual cancer cells and tumor recurrence after liver transplantation. CTC: circulating tumor cells; Damps: danger associated molecular pattern; DBD: donation after brain death; DCD: donation after circulatory death; HOPE: hypothermic oxygenated perfusion; ROS: reactive oxygen species. This figure was designed with the support from biorender.com (accessed on 15 June 2022).

**Table 1 ijms-23-09747-t001:** Studies evaluating risk factors for IRI associated with HCC recurrence.

Study	Data Source	Country	Number of Patients	Donor Type	Preservation	Donor Risk Factors	Recipient Risk Factors	IRI Factors	Outcomes
Croome 2013 [68]	UNOS	USA	5638 vs. 242	DBD vs. DCD	SCS	dWIT, CIT	Age, MELD	NA	DCD liver recipients have a higher HCC recurrence risk compared to DBD recipients; livers with a dWIT of >15 min or a CIT of >6 h 20 min had lower survival rates
Croome 2015 [69]	Single center	USA	340 vs. 57	DBD vs. DCD	SCS	CIT, dWIT (DCD)	AFP, underlying disease severity	NA	HCC recurrence in 12.1% and 12.3% in DBD and DCD liver transplants; good DCD livers have a similar risk of HCC recurrence compared to standard DBD liver recipients
Kornberg 2015 [57]	Single	Germany	106	DBD	SCS(+/− Prostaglandin)	CIT, dWIT	HCC factors	AST/ALT	Up to 23.6% HCC recurrence; prolonged CIT and recipient WIT had higher HCC recurrence. The protective effect of prostaglandin on recurrence-free survival and HCC recurrence more pronounced in recipients outside the Milan criteria
Nagai 2015 [56]	Multi-center	USA	391	DBD	SCS	CIT	WIT, HCC burden, AFP	AST/ALT	15.3% overall recurrence; CIT >10 h and recipient WIT >50 min associated with higher HCC recurrence
Orci 2015 [61]	UNOS	USA	9206 vs. 518	DBD vs. DCD	SCS	dWIT, age, BMI	NA	NA	Donor age >60 y and dWIT were risk factors for increased HCC recurrence
Khorsandi 2015 [70]	Single	UK	256 vs. 91	DBD vs. DCD	SCS	dWIT, CIT	HCC burden	AST, INR	Recipients of good quality DCD livers have similar HCC recurrence risk compared to DBD
Grat 2018 [73]	Single	Poland	195	DBD	SCS	CIT	WIT	AST, LDH, GGT, bilirubin peak, INR	AST ≥1896 U/L increases the risk of HCC recurrence, already in recipients within the Milan criteria
Martinez- Insfran 2019 [71]	Single	Spain	18 vs. 18	DBD vs. DCD	SCS	CIT, dWIT	NA	AST, ALT, prothrombin time	Low risk DCD grafts can be used for standard HCC recipient, with the same recurrence rate compared with transplantation of DBD livers
Silverstein 2020 [8]	UNOS	USA	6996 vs. 567	DBD vs. DCD	SCS	Organ type, dWIT, age, DRI	MELD	NA	Recurrence at 3 y: 7.6% in DCD and 6.4% in DBD livers; DCD livers were an independent predictor of mortality. Donor or graft quality and HCC parameters impact on outcomes

ALT: alanine aminotransferase; ALT: aspartate aminotransferase; DBD: donation after brain death; DCD: donation after circulatory death; DRI: donor risk index; dWIT: donor warm ischemia time; CIT: cold ischemia time; HCC: hepatocellular carcinoma; MELD: model end-stage liver disease; SCS: static cold storage.

**Table 2 ijms-23-09747-t002:** Studies evaluating the impact of MP on IRI-associated HCC recurrence and recipient survival.

Study	Study Type	Country	Number of Patients	Donor Type	Preservation	Donor Risk Factors	Recipient Risk Factors	IRI Factors	Outcomes	Discussion
Mueller 2020 [136]	Multi-center, matched retrospective	UK, Switzerland	70 vs. 70	DBD vs. DCD	HOPE (DCD) vs. SCS (DBD)	Preservation type	HCC burden (DCD HOPE group: 35.7% outside Milan, 28.6% outside UCSF, 18.6% outside Metro-ticket 2.0)	ALT, INR, CRP	HOPE-treated DCD liver recipients had a 5-year tumor-free survival of 92%. 4-fold higher tumor recurrence rate was seen in recipients of unperfused DBD livers compared to DCD grafts with HOPE (25.7% vs. 5.7%, *p* = 0.002)	Retrospective
Tang 2021 [137]	Single center, matched retrospective	China	85 vs. 30	DBD	SCS vs. IFOT	Preservation type	AFP, microvascular invasion	AST, ALT, lactate	Higher recurrence-free survival with IFOT; 1 and 3 y: 92% and 87% IFOT vs. 88% and 53.6% with SCS	Retrospective

AFP: alpha-feto-protein; AST: aspartate aminotransferase; ALT: alanine aminotransferase; DBD: donation after brain death; DCD: donation after circulatory death; CRP: C-reactive protein; HCC: hepatocellular carcinoma; HOPE: hypothermic oxygenated liver perfusion; IFOT: ischemia-free organ transplantation; SCS: static cold storage; UK: United Kingdom.

## Data Availability

Not applicable.

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
