# Peer review of "Mitochondria and Cancer Recurrence after Liver Transplantation—What Is the Benefit of Machine Perfusion?"

_ijms, 2022, doi:10.3390/ijms23179747_

Round 1

Reviewer 1 Report

In this manuscript, Parente et al. reviewed the contributing factors to tumor recurrence after liver transplantation, and discussed the role of mitochondria and ischemia-reperfusion injury associated inflammation in this process. They also summarized the recent findings on machine perfusion technology and its role in liver transplantation. This review is well written and informative. Minor comments and suggestions are shown below.

1. The authors mentioned that ischemia-reperfusion injury induced the release of danger-associated-molecular patters (Damps), which triggers inflammation. It is worthy to discuss what Damps, especially the ones derived from mitochondria, are released, as well as the related pathways, in ischemia-reperfusion injury and liver transplantation.

2. The Tables should be revised to make them more readable and professional. Some of the words in the table were not shown properly. For example, in Table 1, the word “Country” was shown in two lines as “Coun” and “try”. Many words in the tables showed this issue, which made the table hard to read. The font of the words (like the size of the words), and the width of the cells should be adjusted.  

3. Though the manuscript is well written, there are some typos. For example, in line 127, “Orgen” should be “Organ”. In line 268, should the “AFP” here be “ATP”? If it is “AFP”, its full name should be indicated here, but not in the next paragraph (Line 280). Such typos should be checked carefully throughout the manuscript.

Author Response

Reviewer #1

In this manuscript, Parente et al. reviewed the contributing factors to tumor recurrence after liver transplantation, and discussed the role of mitochondria and ischemia-reperfusion injury associated inflammation in this process. They also summarized the recent findings on machine perfusion technology and its role in liver transplantation. This review is well written and informative. Minor comments and suggestions are shown below.

Reply: We thank the reviewer for the kind evaluation of our manuscript and the suggestions and comments.

  1. The authors mentioned that ischemia-reperfusion injury induced the release of danger-associated-molecular patters (Damps), which triggers inflammation. It is worthy to discuss what Damps, especially the ones derived from mitochondria, are released, as well as the related pathways, in ischemia-reperfusion injury and liver transplantation.

Reply: We thank the reviewer for this important point. A new subparagraph has been added to the section “Mechanisms of ischemia-reperfusion injury” (please see page 3 of the revised manuscript), which provides more details regarding molecular pathways associated with ROS and various DAMPS molecules released.

  1. The Tables should be revised to make them more readable and professional. Some of the words in the table were not shown properly. For example, in Table 1, the word “Country” was shown in two lines as “Coun” and “try”. Many words in the tables showed this issue, which made the table hard to read. The font of the words (like the size of the words), and the width of the cells should be adjusted.  

Reply: We thank the reviewer for highlighting this point. Table’s font and size have been revised for an improved layout and readability. Please see pages 8 and 13.  

  1. Though the manuscript is well written, there are some typos. For example, in line 127, “Orgen” should be “Organ”. In line 268, should the “AFP” here be “ATP”? If it is “AFP”, its full name should be indicated here, but not in the next paragraph (Line 280). Such typos should be checked carefully throughout the manuscript.

Reply: We thank the reviewer for this important comment. In line 268 AFP is alpha feto protein and we have added this abbreviation here and also in line 280. The manuscript underwent major revisions and was checked for all sorts of typos with subsequent highlighted changes.

Reviewer 2 Report

Parente et al. reviewed underlying mechanisms of mitochondria and cancer recurrence after liver transplantation (LT). The manuscript is well-written and well-organized. However, I have a few concerns regarding the manuscript. The manuscript may improve if it addresses the concerns listed below.

(1) As for "cancer", did the authors limited cancer to HCC? or included others?

(2) A number of HCCs develop with underlying cirrhotic liver and/or HBV/HCV. Is the recipient's condition (before LT) associated with outcomes after LT? what is the association of underlying liver diseases with mitochondria and machine perfusion?

(3) what is the already known or expected association of mitochondria and machine perfusion with metastasis, such as pulmonary metastasis? 

(4) As for the potential impact of IRI on the recurrence of other liver tumors, please explain the specific mechanism of this with iCCA since LT for iCCA has been excluded from the contraindications in a number of LT centers.

(5) The depth of the underlying biological mechanisms is not deep throughout the manuscript. For example, as for genetic mutations, which gene and loci?

Author Response

Reviewer #2

Parente et al. reviewed underlying mechanisms of mitochondria and cancer recurrence after liver transplantation (LT). The manuscript is well-written and well-organized. However, I have a few concerns regarding the manuscript. The manuscript may improve if it addresses the concerns listed below.

Reply: We thank the reviewer for the kind evaluation of the manuscript. The suggestions and comments were very well received and are addressed below.

(1) As for "cancer", did the authors limited cancer to HCC? or included others?

Reply: We thank the reviewer for this important question. The manuscript was mainly focused on HCC and related recurrence based on the largest body of literature available for this type of liver cancer; equally there is clinical evidence for the protective effect of machine perfusion (Mueller Ann Surg 2020 and Tang 2021). However, given the recent advancement in the field of transplant oncology with an increasing number of candidates accepted for liver transplantation with other liver cancer types and metastases, we have included other liver tumours and metastases in the last main paragraph (no.7) on page 13. We have also added more insights regarding underlying mechanisms related to colorectal liver metastasis and cholangiocarcinoma; this modification was also done to address the reviewer’s comment #6.

(2) A number of HCCs develop with underlying cirrhotic liver and/or HBV/HCV. Is the recipient's condition (before LT) associated with outcomes after LT? what is the association of underlying liver diseases with mitochondria and machine perfusion?

Reply: This is a very important point. The underlying liver disease might well contribute to cancer recurrence after transplantation. Despite the large body of evidence for the link between HCC occurrence and a chronic inflammation based on hepatitis B or C and for example non-alcoholic fatty liver disease (NAFLD), the data for this link after liver transplantation is scarce. More research is required to better understand how the underlying liver disease could impact on tumour recurrence with and without machine perfusion. This has been highlighted in the conclusions as a potential field for future research.

(3) what is the already known or expected association of mitochondria and machine perfusion with metastasis, such as pulmonary metastasis? 

Reply: We thank the reviewer for this important point. Liver candidates with HCC and other cancers are listed and transplanted while their tumour burden is limited to the deceased liver, which is removed during transplantation. Based on the manipulation and medical/anesthesiological treatment during surgery, a potentially higher number of tumor cells circulate in the body. When a new liver is transplanted, the ischemia reperfusion injury creates a proinflammatory environment attractive for circulating cancer cells to resettle and regrow into a local recurrence found in the newly implanted liver. An elevated ischemia reperfusion injury is an inflammatory status, which effects the entire body, up to an extreme of severe initial liver dysfunction or even primary non function. This status of elevated reperfusion injury is well comparable with a systemic inflammatory response, and frequently combined with acute kidney injury and lung injury during the initial period after liver transplantation, where the recipients body tries to cope with the injury. This induces a similar inflammation in other organs, e.g., the lung, as seen in the new liver (local IRI, microenvironmental changes). Circulating tumor cells were found to easier resettle and grow in such proinflammatory tissues, including the lung. This is further supported by the anatomical position of the lung directly after liver and heart with a large capillary network of small diameter vessels, surrounding the alveoli. Mitochondria are the main instigators of reperfusion injury and trigger such downstream inflammation. To maintain mitochondria healthy is therefore the key to reduce and prevent IRI, inflammation and cancer recurrence. While interstitial macrophages seem to play a key role, further studies are needed for a better understanding of the entire mechanism. The link of release of damps and cytokines after liver transplantation with later dysfunction of recipient kidneys was described and more research is required for lungs and other organs (e.g., PMID: 35272878).  

(4) As for the potential impact of IRI on the recurrence of other liver tumors, please explain the specific mechanism of this with iCCA since LT for iCCA has been excluded from the contraindications in a number of LT centers.

Reply: We thank the reviewer; this is a truly important point. We have added further details on CCA; please see also response to comment #1 and the revised manuscript on page 14 (revised chapter 7).

(5) The depth of the underlying biological mechanisms is not deep throughout the manuscript. For example, as for genetic mutations, which gene and loci?

Reply: We thank the reviewer for this important point. With our manuscript we have mainly focussed on the recurrence of liver cancers after transplantation, which is linked to a resettling of already existing tumour cells, which hide or circulate and subsequently reattach to inflamed endothelial cells in a reperfusion-injury-affected microenvironment of a newly implanted organ. In contrast, the occurrence of new cancers (e.g., through genetic mutations of nuclear and mitochondrial) as a response to injury and an imbalance between reperfusion injury, immune response and medical immunosuppression, follows different mechanisms, and is affected by injured mitochondria. While as such the level of injury and mitochondria triggering genetic mutations is key to develop new cancers from healthy liver tissues (or other cells), the recurrence is linked to already existing tumor cells and follows a different mechanism as previously described. We have added a few lines regarding such differences in the summary chapter. Please see page 14/15 of the revised manuscript.

Round 2

Reviewer 2 Report

The authors have made modifications for the raised concerns. I have no more concerns.